# ROBUSTNESS GUARANTEES FOR ADVERSARIAL TRAINING ON NON-SEPARABLE DATA

## ABSTRACT

Adversarial training has emerged as a popular approach for training models that are robust to inference time attacks. However, our theoretical understanding of why and when it works remains limited. Prior work has offered convergence analysis of adversarial training, but they are either restricted to the Neural Tangent Kernel (NTK) regime or make restrictive assumptions about data such as linearly realizability. In this work, we provide convergence and generalization guarantees for adversarial training of two-layer networks of any width on non-separable data. Our analysis goes beyond the NTK regime and holds for both smooth and non-smooth activation functions. We support our theoretical findings with an empirical study on synthetic and real-world data.

## 1 INTRODUCTION

Machine learning models are ubiquitous in real-world applications, achieving state-of-the-art performance on various tasks such as image classification and speech recognition. However, several recent studies have shown that these models, especially those based on deep neural networks, are highly vulnerable to small, nearly imperceptible, albeit strategic, perturbation of data. These perturbations, called adversarial examples, are abundant and easy to find computationally (Bubeck et al., 2021; Wang et al., 2022). The potential of such adversarial attacks to substantially degrade the performance of an otherwise well-performing model has been a source of significant concern regarding deploying machine learning models in real-world systems. It is no surprise, then, that developing algorithms that can provably defend against such attacks and are guaranteed to improve the robustness of machine learning has gained tremendous traction in recent years.

One of the most prominent empirical defense algorithms against inference-time attacks is the adversarial training method of Madry et al. (2018). Adversarial training proceeds by simulating attacks as part of training – generating adversarial examples from (clean) training examples and using them to train a neural network. We can view adversarial training as a two-player game, wherein the learner seeks to minimize their error on the training set while an adversary strives to maximize the error by crafting small strategic corruptions of the input training examples. Several empirical studies show that by using adversarial training, the learner returns a model that is more resilient to perturbations in the input space (Madry et al., 2018; Shafahi et al., 2019b; Dong et al., 2020; Pang et al., 2021).

Despite the empirical success of adversarial training, our understanding of its theoretical underpinnings is far from complete. Several prior works study statistical and computational aspects of adversarial training but in somewhat restrictive settings; e.g., assuming linear separability of data (Mianjy and Arora, 2022), or essentially assuming away nonconvexity of neural networks by considering an overly parametrized regime wherein the trajectory dynamics are in the lazy regime (aka, the neural tangent kernel or the NTK setting) (Gao et al., 2019; Zhang et al., 2020; Li and Telgarsky, 2023). In this paper, we forego these simplifying assumptions and present theoretical convergence and generalization guarantees for adversarial training on two-layer neural networks, of any width, on non-separable data. Our key contributions are as follows.

1. We establish convergence guarantees for adversarial training of two-layer neural networks. We allow the network to be of *arbitrary width* thereby extending our results to networks beyond the NTK regime. Furthermore, we do not make any assumptions about the separability or robust realizability of data.

2. We provide generalization guarantees on both the clean test error and the robust test error. For a moderately large network, we show that for norm-bounded additive adversarial attacks, if the perturbation budget is not too large, the robust test error approximates the label noise rate. For adversarial attacks with a large perturbation budget, we show that the robust test error is bounded from below by a constant.

3. We validate our theoretical results with experiments on both synthetic and real-world datasets.

## 1.1 RELATED WORK

**Convergence Analysis of Standard Training.** Several recent works study the convergence of (stochastic) gradient descent for training neural networks (Arora et al., 2019; Allen-Zhu et al., 2019; Cao and Gu, 2019). Most of these works focus on a lazy training regime wherein the network weights remain close to initialization through the run of the algorithm (owing to an extreme over-parametrization); this is also referred to as the neural tangent kernel (NTK) setting. While interesting from a theoretical perspective (we essentially end up with a convex learning problem), this assumption is typically violated in practice. Analyzing SGD beyond the NTK setting is much more challenging owing to the non-convexity of learning problems associated with training neural networks of arbitrary widths. There has been some progress toward addressing this challenge – Frei et al. (2022) provide a first guarantee for finite-width neural networks trained on logistic loss for data drawn from a Gaussian mixture model. Concurrently, Cao et al. (2022) characterize the generalization guarantees of two-layer convolutional neural networks, assuming that the input data is a sum of a label-dependent signal patch and a label-independent noise patch. While both of the works above consider a smooth activation function, follow-up works by Kou et al. (2023); Xu and Gu (2023) extend the result to SGD for training neural networks with non-smooth activation functions (e.g., ReLU networks).

**Convergence Analysis of Adversarial Training.** Adversarial training, introduced by Madry et al. (2018), is one of the most popular algorithms for training models that are robust to adversarial attacks. Subsequent works have explored variants, including the TRADES (Zhang et al., 2019) and MART (Wang et al., 2020) algorithms. Despite their success, a theoretical understanding of why and when adversarial training succeeds remains elusive. Much of the recent work (Charles et al., 2019; Li et al., 2020; Zou et al., 2021; Chen et al., 2021) has focused on studying adversarial training of linear models wherein the adversarial examples are given in a simple closed-form expression – this simplifies the problem greatly reducing it to standard training. Adversarial training of neural networks was analyzed by Gao et al. (2019) and further improved by Zhang et al. (2020); however, both of these works focus on ensuring convergence of the training procedure and do not provide generalization guarantees on robust loss. This gap has been addressed in very recent work by Li and Telgarsky (2023). However, the work of Li and Telgarsky (2023), and the prior work all focus on the lazy training regime, which, unfortunately, has been proven to be at odds with robustness Wang et al. (2022). Finally, Mianjy and Arora (2022) provide an end-to-end analysis of adversarial training beyond the NTK setting with a variant of adversarial training that involves using a slightly different (reflected) loss for the inner loop maximization problem (for finding an attack vector as part of adversarial training). The results of Mianjy and Arora (2022) are limited to distributions that are robustly realizable.

Our work builds on that of Frei et al. (2022) and considers a high-dimensional setting for a class-conditional model; the data model, as well as various other data assumptions we need, were first introduced and studied in Chatterji and Long (2021). While our proof techniques are inspired by Frei et al. (2022), we differ in many respects. To the best of our knowledge, ours is the first work that provides the convergence and generalization guarantees for *adversarial training* for a *non-separable data distribution*. We consider neural networks with both smooth and non-smooth activation functions, e.g., ReLU networks; the analysis of Frei et al. (2022) is limited to smooth activation functions. Additionally, unlike prior works (Gao et al., 2019; Zhang et al., 2020; Li and Telgarsky, 2023) that are limited to the NTK setting, our guarantees hold for neural networks of *arbitrary width* and analyze GD-based adversarial training in the rich regime (i.e., *beyond the lazy regime*).

## 2 PRELIMINARIES

**Notation** Throughout the paper, we denote scalars, vectors, and matrices with lowercase italics, lowercase bold, and uppercase bold Roman letters, respectively; e.g., $u$, u, and U. We use $[m]$

to denote the set $\{1, 2, \ldots, m\}$ and use both $\|\cdot\|$ and $\|\cdot\|_2$ for $\ell_2$-norm. Given a matrix $U = [u_1, \ldots, u_m] \in \mathbb{R}^{d \times m}$, we use $\|U\|_F$ and $\|U\|_2$ to represent the Frobenius norm and spectral norm, respectively. We use $\mathcal{B}_2(u, \alpha)$ to denote the $\ell_2$ ball centered at $u \in \mathbb{R}^d$ of radius $\alpha$. We use the standard O-notation ($\mathcal{O}$, $\Theta$ and $\Omega$).

## 2.1 PROBLEM SETUP

We focus on binary classification and denote the input space and label space as $\mathcal{X} = \mathbb{R}^d, \mathcal{Y} = \{\pm 1\}$, respectively. We assume that the data are drawn from a noisy mixture data distribution $\mathcal{D}$ on $\mathcal{X} \times \mathcal{Y}$ that, along with its variants, has been studied in several recent works (Chatterji and Long, 2021; Cao et al., 2021; Frei et al., 2022). Formally, we consider the following data distribution.

**Definition** (Data Distribution). Let $\mathcal{D}_{\text{clust}}$ be a $\lambda$-strongly log-concave distribution over $\mathbb{R}^d$ for some $\lambda > 0$. We assume that $\mathcal{D}_{\text{clust}} = \mathcal{D}_{\text{clust}}^{(1)} \times \cdots \times \mathcal{D}_{\text{clust}}^{(d)}$ is a product distribution whose marginals are all mean-zero with the sub-Gaussian norm at most one. We further assume that $\mathbb{E}_{\xi \sim \mathcal{D}_{\text{clust}}}[\|\xi\|^2] \geq \kappa d$ holds for some $0 < \kappa < 1$. Let $\mathcal{D}_c$ be a distribution over $\mathcal{X} \times \mathcal{Y}$. We first draw a sample $(x_c, y_c) \sim \mathcal{D}_c$ by sampling $y_c \in \{\pm 1\}$ uniformly at random, sampling $\xi \sim \mathcal{D}_{\text{clust}}$, and setting $x_c = y_c \mu + \xi$. Given a noise rate $\beta > 0$, we define our true data distribution $\mathcal{D}$ to be any distribution over $\mathcal{X} \times \mathcal{Y}$ such that the marginal distribution of $\mathcal{D}$ and $\mathcal{D}_c$ on $\mathcal{X}$ are the same, and the total variation distance between the two distributions is bounded by $\beta$, i.e., $d_{\text{TV}}(\mathcal{D}_c, \mathcal{D}) \leq \beta$.

The standard coupling lemma states that given two distributions $\mathcal{D}$ and $\mathcal{D}_c$ over the same domain $\mathcal{Z} = \mathcal{X} \times \mathcal{Y}$, there exists a joint distribution over $\mathcal{Z} \times \mathcal{Z}$ such that the marginals along the projections $(z, z') \mapsto z$ and $(z, z') \mapsto z'$ are $\mathcal{D}$ and $\mathcal{D}_c$, respectively. Given that the marginal on $\mathcal{X}$ for $\mathcal{D}$ and $\mathcal{D}_c$ are the same (see the definition above), this implies that for $(x, y) \sim \mathcal{D}, (x_c, y_c) \sim \mathcal{D}_c, P(x = x_c) = 1$ and $P(y \neq y_c) \leq \beta$. The definition above includes two settings: 1) Independent label flip, where for each sample, label $y$ is obtained by flipping $y_c$ with probability at most $\beta$, independent of how other labels are generated; 2) Non-independent label flip, where there exists potential correlations between labels $y$. A yet another special instance that has been studied extensively in the adversarial learning literature is that of Gaussian distribution (Javanmard et al., 2020; Dobriban et al., 2020; Dan et al., 2020) which is a special case of the data generative model above for $\beta = 0$.

**Hypothesis Class.** We focus on learning two-layer neural networks defined as: $f(x; W) := \frac{1}{\sqrt{m}} \sum_{s=1}^m a_s \phi(\langle w_s, x \rangle)$ where $m$ is an even integer representing the number of hidden nodes and $\phi : \mathbb{R} \to \mathbb{R}$ is an activation function. The weight matrix at the bottom layer is denoted as $W = [w_1, \ldots, w_m] \in \mathbb{R}^{d \times m}$ and the weight vector at the top layer by $a = [a_1, \ldots, a_m] = [1, \ldots, 1, -1, \ldots, -1] \in \mathbb{R}^m$. The top layer weight vector $a$ is kept fixed throughout the training process. The weight vectors at the bottom layer are initialized randomly as $w_s^0 \sim N(0, \omega_{\text{init}}^2 I)$, for $s \in \{1, \ldots, \frac{m}{2}\}$, and setting $w_s^0 = w_{s-\frac{m}{2}}^0$ for $s \in \{\frac{m}{2} + 1, \ldots, m\}$. This ensures symmetry at initialization and yields $f(x; W^0) = 0$ for all x. This symmetric initialization technique is commonly used in related work (Langer, 2021; Bartlett et al., 2021; Montanari and Zhong, 2022) and we employ here for analytical purposes.

**Training Data.** We are given a training data of size $n$ sampled i.i.d. from the noisy data distribution, $\mathcal{S} = \{(x_i, y_i)\}_{i=1}^n \sim \mathcal{D}$. Let $\mathcal{C}$ denote the set of indices of training data corresponding to the clean labels; i.e., for $i \in \mathcal{C}$, we have that $(x_i, y_i) \sim \mathcal{D}_c$; similarly, let $\mathcal{N}$ to denote the set of indices corresponding to noisy labels; i.e., $(x_i, -y_i) \sim \mathcal{D}_c$ for all $i \in \mathcal{N}$.

**Loss Function.** The 0-1 loss of a predictor $f(\cdot, W)$ on a data point $(x, y)$ is defined as $\ell^{0/1}((x, y); W) = \mathbb{1}(yf(x; W) \leq 0)$, where $\mathbb{1}(\cdot)$ is the indicator function. For computational reasons, as is typical, we use the logistic loss, denoted $\ell(z) = \log(1 + \exp(-z))$, to train the two-layer neural networks. The population and the empirical loss w.r.t. $\ell(\cdot)$ are denoted as:

$$L(W) := \mathbb{E}_{(x,y) \sim \mathcal{D}} \ell(yf(x; W)), \text{ and } \widehat{L}(W) := \frac{1}{n} \sum_{i=1}^n \ell(y_i f(x_i; W)).$$

**Robust Loss.** We consider $\ell_2$ norm-bounded adversarial attacks with a perturbation budget of size $\alpha > 0$. The set of all such perturbations for an input example $x \in \mathcal{X}$ is represented by $\mathcal{B}_2(x, \alpha)$. This threat model motivates minimizing the robust 0-1 loss defined as $\ell_{\text{rob}}^{0/1}((x, y); W) =$

$\max_{\tilde{x} \in \mathcal{B}_2(x,\alpha)} \mathbb{1}(yf(\tilde{x}; W) \leq 0)$. The population and empirical risk w.r.t. the 0-1 loss and the robust 0-1 loss, respectively, are denoted as $L^{0/1}$, $\widehat{L}^{0/1}$, $L_{\text{rob}}^{0/1}$, and $\widehat{L}_{\text{rob}}^{0/1}$. Analogously, the population and empirical risk w.r.t. the (surrogate) logistic loss $\ell(\cdot)$ are defined as:

$$L_{\text{rob}}(W) := \mathbb{E}_{(x,y) \sim \mathcal{D}} \max_{\tilde{x} \in \mathcal{B}_2(x,\alpha)} \ell(yf(\tilde{x}; W)), \text{ and } \widehat{L}_{\text{rob}}(W) := \frac{1}{n} \sum_{i=1}^{n} \max_{\tilde{x}_i \in \mathcal{B}_2(x_i,\alpha)} \ell(y_i f(\tilde{x}_i; W)).$$

Note that we are ultimately interested in bounding the 0-1 loss and its robust variant.

**Adversarial Training.** The gradient descent-based adversarial training algorithm is presented in Algorithm 1. We denote the adversarial training example for some input $x_i$ given model parameter $W^t$, at round $t$ as $\tilde{x}_i^t = \arg\max_{\tilde{x}_i \in \mathcal{B}_2(x_i,\alpha)} \ell(y_i f(\tilde{x}_i; W^t)) = \arg\min_{\tilde{x}_i \in \mathcal{B}_2(x_i,\alpha)} y_i f(\tilde{x}_i; W^t)$.

---

**Algorithm 1** Gradient Descent-based Adversarial Training

**Input:** Step size $\eta$, perturbation budget per sample $\alpha$. Number of iterations $T$.
1: Initialize $W^0$ randomly.
2: **for** $t = 0, \ldots, T-1$ **do**
3:     **for** $i = 1, \ldots, n$ **do**
4:         $\tilde{x}_i^t = \arg\max_{\tilde{x}_i \in \mathcal{B}_2(x_i,\alpha)} \ell(y_i f(\tilde{x}_i; W^t))$.
5:     **end for**
6:     Update $W^{t+1} = W^t - \frac{\eta}{n} \sum_{i=1}^{n} \nabla \ell(y_i f(\tilde{x}_i^t; W^t))$
7: **end for**
8: return: $W^T$

---

## 3 MAIN RESULT

### 3.1 SMOOTH ACTIVATION FUNCTION

In this section, we consider a strictly increasing, 1-Lipschitz, $H$-smooth activation function that is approximately homogeneous with $\phi(0) = 0$. Formally, there exists $\gamma, H > 0, 0 \leq \zeta < 1, c_1 \geq 0, c_2 \geq 0$ such that

$$0 < \gamma \leq \phi'(z) \leq 1, \phi'(z) \text{ is } H\text{-Lipschitz}, \text{ and } |\phi'(z) \cdot z - \phi(z)| \leq c_1 + c_2 |z|^\zeta, \forall z \in \mathbb{R}.$$

Smooth activation functions have been extensively studied both theoretically and empirically (Liu and Di, 2021; Biswas et al., 2022). One example of such an activation function that satisfies our condition is the smoothed Leaky ReLU activation (Frei et al., 2022) defined as follows:

$$\phi_{\text{SLReLU}}(z) = \begin{cases} z - \frac{1-\gamma}{4H}, & z \geq \frac{1}{H} \\ \frac{1-\gamma}{4}Hz^2 + \frac{1+\gamma}{2}z, & |z| \leq \frac{1}{H} \\ \gamma z - \frac{1-\gamma}{4H}, & z \leq -\frac{1}{H} \end{cases} . \tag{1}$$

However, we do need an additional assumption on top of what Frei et al. (2022) require. In particular, we assume that $\phi'(z)z$ and $\phi(z)$ are close to each other. We argue that this is a mild assumption, and holds trivially for standard ReLU and Leaky ReLU, with $c_1 = c_2 = 0$. For $\phi_{\text{SLReLU}}(z)$, of Frei et al. (2022), the assumption holds with $\zeta = 0$ with $c_1 = \frac{1-\gamma}{4H}$, and $c_2 = 0$. The reason we need this additional assumption is because the neural networks with $\phi_{\text{SLReLU}}(z)$ activation function are no longer homogeneous. Consequently, without the assumption we end up with terms in the upper bound on the empirical robust risk that depends on the Frobenius norm of the weight matrix (see Section 4.2 for more details).

We make the following set of assumptions about our problem setup. Specifically, we consider a high dimensional setting where the dimension $d$ is much larger than the number of training samples $n$, as stated below in Assumption (A2). Such a regime is popular in biomedical settings where the data comes from limited patient information such as MRI or DNA sequence. Assumption (A6) requires a small initialization to ensure that the first step of adversarial training dominates the behavior of the neural network, pushing it beyond the lazy training regime. Such initialization technique has also been introduced in previous work (Ba et al., 2019; Xing et al., 2021). Given that the objective of adversarial training is to achieve a classifier that is robust against small input perturbations imperceptible to human eyes, Assumption (A7) is reasonable as it imposes a mild constraint on the attack strength. Finally, we note that when $\alpha = 0$, these assumptions are essentially the same as in Frei et al. (2022).

**Assumption 1.** Let $\delta \in (0, 1/2)$. We assume that there exists a positive constant $C$ such that the following holds: (A1) The number of samples satisfies $n > C \log(1/\delta)$. (A2) The dimension satisfies $C \max\{\|\mu\|^2 n, n^2 (\log(n/\delta) + \alpha^2)\} \leq d \leq \|\mu\|^4 / C$. (A3) The signal size satisfies $\|\mu\|^2 \geq C \log(n/\delta)$. (A4) noise rate $\beta \in [0, 1/C]$. (A5) Step size $\eta \leq (Cd^2(1 + \frac{H}{\sqrt{m}})^2)^{-1}$. (A6) Initialization variance satisfies $\omega_{\text{init}}\sqrt{md} \leq \eta$. (A7) Adversarial perturbation $\alpha \leq 0.99 \|\mu\|$.

Next we present our main result of this section that describes the effects of adversarial training on a neural network with smooth activation functions trained on samples from the noisy distribution $\mathcal{D}$ (see Section 2.1). Our findings suggest that, as the we run adversarial training for more epochs, the robust training loss goes to zero. Furthermore, the clean test error and the robust test error is approximately equal to the noise rate, provided that the attack strength, $\alpha$, is small.

**Theorem 3.1.** Let $0 < \varepsilon \leq \frac{1}{2n}, \delta \in (0, 1/2)$. Let $\phi$ be a $\gamma$-leaky $H$-smooth activation with $0 \leq \zeta < 1$. Let $\kappa \in (0, 1), \lambda > 0$. Then, given that Assumption 1 holds with some constant $C > 0$, there exists a constant $c > 0$ such that after running Algorithm 1 for $T \geq \Omega\left(\left(\frac{1+\sqrt{m/d^3}}{(199\|\mu\|-200\alpha)\gamma\eta\varepsilon}\right)^{\frac{2}{1-\zeta}}\right)$ iterations, we have that with probability at least $1 - 2\delta$ over the random initialization and the draw of an i.i.d. sample of size $n$, the following holds:

1. The robust training loss satisfies $\widehat{L}_{\mathrm{rob}}(\mathbf{W}^T) \leq \varepsilon$.

2. The clean test error satisfies $L^{0/1}(\mathbf{W}^T) \leq \beta + 2\exp\left(-\frac{c\lambda n\|\mu\|^4}{C^2 d}\left(0.99 - \frac{\alpha}{\|\mu\|}\right)^2\right)$.

3. For $\frac{\alpha}{\|\mu\|} \leq \frac{0.99\sqrt{n}\|\mu\|}{\sqrt{n}\|\mu\|+C\sqrt{d}}$, the robust test error satisfies

$$L_{\mathrm{rob}}^{0/1}(\mathbf{W}^T) \leq \beta + 2\exp\left(-c\lambda\|\mu\|^2\left(\frac{\sqrt{n}\|\mu\|}{C\sqrt{d}}\left(0.99 - \frac{\alpha}{\|\mu\|}\right) - \frac{\alpha}{\|\mu\|}\right)^2\right).$$

For the smooth Leaky ReLU activation function of Frei et al. (2022), we have the following result.

**Corollary 3.2.** For any $\gamma$-leaky $H$-smooth ReLU activation $\phi_{\mathrm{SLReLU}}$ defined in Equation (1), and for all $\kappa \in (0, 1), \lambda > 0$, given Assumption 1 holds, we have that with probability at least $1 - 2\delta$ over the random initialization and the draws of the samples, the robust training loss satisfies

$$\widehat{L}_{\mathrm{rob}}(\mathbf{W}^T) \leq \mathcal{O}\left(\frac{1 + \sqrt{(1-\gamma)/H}m^{1/4}}{(199\|\mu\| - 200\alpha)\gamma\sqrt{\eta}\sqrt{T}}\right).$$

## 3.2 NON-SMOOTH ACTIVATION FUNCTION

Here, we consider a more practical setting where the activation function is no longer smooth. We consider a homogeneous non-smooth activation function that satisfies the following properties.

$$\phi(0) = 0, \phi'(z)z = \phi(z), z \in \mathbb{R}; \quad 0 \leq \phi'(z) \leq 1, z \in \mathbb{R}; \quad \phi'(z) \geq \gamma, z \geq 0, \gamma \in (0, 1].$$

This includes ReLU and Leaky ReLU activation functions. Additionally, we assume the following.

**Assumption 2.** Let $\delta \in (0, 1/2)$. We assume that there exists a positive constant $C$ such that the following holds: (B1) The network width satisfies $m \geq C\log(n/\delta)$. (B2) The signal size satisfies $\|\mu\| \geq C\max\left\{\left(\frac{d}{n}\log(md/n\delta)\right)^{1/4}, \sqrt{\log(n/\delta)}\right\}$. (B3) The dimension satisfies $d \geq C\max\{\|\mu\|^2 n, n^2(\log(n/\delta) + \alpha^2)\}$. (B4) noise rate $\beta \in [0, 1/C]$. (B5) Initialization variance satisfies $\omega_{\mathrm{init}}\sqrt{md} \leq \eta$. (B6) Step size $\eta \leq (Cd^2)^{-1}$. (B7) The number of samples satisfies $n \geq C\log(m/\delta)$. (B8) Adversarial perturbation $\alpha \leq \sqrt{n/d}\|\mu\|$.

Assumption (B1) is a relatively mild constraint on the network width. Assumption (B2) is slightly more stringent compared to Assumption (A3). However, it is worth noting that in the clean setting, the minimax generalization error is at least $\mathcal{O}\left(\exp\left(-\min\left(\|\mu\|^2, n\|\mu\|^4/d\right)\right)\right)$ (Giraud and Verzelen, 2019), implying that Assumption(B2) is unavoidable up to logarithmic factors if we desire a classifier with good generalization. Assumptions (B7) and (B8) are also more restrictive compared to Assumptions (A1) and (A7), respectively. These assumptions ensure the presence of sufficient number of neurons to have positive activation at the initial stage of adversarial training, which is a crucial aspect of our analysis in terms of relaxing the requirement of a smooth activation function, as opposed to Section 3.1. The analogous result to Theorem 3.1 is presented below.

**Theorem 3.3.** Let $0 < \varepsilon \leq \frac{1}{2n}, \delta \in (0, 1/2)$. Let $\phi$ be a non-smooth activation with $\gamma \in (0, 1]$. Let $\kappa \in (0, 1), \lambda > 0$. Then, given that Assumption 2 holds with some constant $C > 0$, there exists a constant $c > 0$ such that after running Algorithm 1 for $T \geq \Omega\left(\left((199\|\mu\| - 200\alpha)\gamma\sqrt{\eta}\varepsilon\right)^{-2}\right)$ iterations, we have that with probability at least $1 - 2\delta$ over the random initialization and the draw of an i.i.d. sample of size $n$, the following holds:

1. The robust training loss satisfies $\widehat{L}_{\text{rob}}(\mathbf{W}^T) \le \varepsilon$.
2. The clean test error satisfies $L^{0/1}(\mathbf{W}^T) \le \beta + 2\exp\left(-\frac{c\lambda n \|\mu\|^4}{C^2 d}\left(1 - \frac{\alpha}{\|\mu\|}\right)^2\right)$.
3. For $\frac{\alpha}{\|\mu\|} \le \frac{\sqrt{n}\|\mu\|}{\sqrt{n}\|\mu\| + C\sqrt{d}}$, the robust test error satisfies

$$L_{\text{rob}}^{0/1}(\mathbf{W}^T) \le \beta + 2\exp\left(-c\lambda\|\mu\|^2\left(\frac{\sqrt{n}\|\mu\|}{C\sqrt{d}}\left(1 - \frac{\alpha}{\|\mu\|}\right) - \frac{\alpha}{\|\mu\|}\right)^2\right).$$

### 3.3 DISCUSSION

Theorems 3.1 and 3.3 suggest an interesting interplay between the parameters $d$, $n$, and $\|\mu\|$ as described in Assumptions 1 and 2. Importantly, when $n \ge \tilde{\Omega}\left(\frac{d\max(1,\alpha^2)}{\|\mu\|^2(\|\mu\|-\alpha)^2}\right)$, it ensures a small robust test error. Furthermore, when $n \ge \tilde{\Omega}\left(\frac{d}{\|\mu\|^2(\|\mu\|-\alpha)^2}\right)$, the clean test error is also guaranteed to be small. In cases where $\alpha = 0$, Theorem 3.1 and 3.3 recover the results in the standard setting (Frei et al., 2022; Xu and Gu, 2023). Compared to the standard setting, we pay an additional price proportional to $\frac{\max(1,\alpha^2)}{(1-\alpha/\|\mu\|)^2}$ in terms of the sample size. It is worth noting that both the clean test error and the robust test error decrease as $n/d$ increases or the attack strength $\frac{\alpha}{\|\mu\|}$ decreases, which is consistent with the findings in previous literature (Schmidt et al., 2018; Shafahi et al., 2019a).

Next, we provide a lower bound on the robust test error that is independent of the algorithm as well as the hypothesis class.

**Theorem 3.4.** We consider independent label flip with probability $\beta$. Let $p(\mathrm{x})$ be the density function of $\mathcal{D}_{\text{clust}}$. For any given classifier $f(\cdot;\mathbf{W})$, when $\alpha < \|\mu\|$, we have $L_{\text{rob}}^{0/1}(\mathbf{W}) \ge \beta + \frac{1-2\beta}{4}\int_{\mathbb{R}^d}\min\{p(\xi), p(\xi+\mathrm{v})\}d\xi$, where $\mathrm{v} = 2\left(1 - \alpha/\|\mu\|\right)\mu$. When $\alpha \ge \|\mu\|$, the robust test error satisfies $L_{\text{rob}}^{0/1}(\mathbf{W}) \ge 0.5$.

Consider the special instance of when $\mathcal{D}_{\text{clust}}$ is a standard Gaussian distribution. Theorem 3.4 recovers the optimal risk in Dobriban et al. (2020) up to a scaling factor when $\beta = 0$. Moreover, the upper bound on the robust test error (denoted as UBD) that we provide in Theorems 3.1 and 3.3 and the lower bound (denoted as LBD) in 3.4 satisfy the following relationship: $(\text{UBD}-\beta) = (\text{LBD}-\beta)^{\mathcal{O}(n\|\mu\|^2/d)}$. When $\frac{n\|\mu\|^2}{d} = \Omega(1), \alpha \le \mathcal{O}(\|\mu\|)$, our upper bound roughly matches the lower bound.

**Overfitting with Adversarial Training.** Recent empirical studies have observed overfitting with adversarial training, wherein the robust training loss continues to decrease with the number of epochs, whereas the robust test error first decreases and then starts increasing (Rice et al., 2020). While our result may, at first, seem in conflict with this empirical observation, we note that there is actually no contradiction since we consider a specific data-generative model and a bound on the size of the adversarial perturbation during adversarial training. Indeed, recent empirical studies by Dong et al. (2021) and Yu et al. (2022) confirm that small $\alpha$ prevents adversarial training from overfitting. Furthermore, Xing et al. (2022) explored the phase transition between standard training and adversarial training and showed that the optimization trajectories in the two settings are close to each other when $\alpha$ is small. One interesting future direction is to justify the generalization guarantee for moderately large attack strength $\frac{\alpha}{\|\mu\|}$.

**Comparison with Theoretical Works** Several recent works focus on giving convergence and generalization guarantees for adversarial training (Gao et al., 2019; Zhang et al., 2020; Mianjy and Arora, 2022; Li and Telgarsky, 2023); here we compare and contrast our work with each of these.

The work of Gao et al. (2019) prove convergence for a modified algorithm for adversarial training wherein the iterates are projected onto a norm ball to ensure that the network weights stay close to initialization. However, they further need to assume that a robust network exists in the vicinity of the initialization. Such an assumption has been shown to be invalid in a recent work (Wang et al., 2022). In a related work, Zhang et al. (2020) provide a fine-grained convergence analysis for datasets that are well-separated. More recently, Li and Telgarsky (2023) give convergence and generalization guarantees for adversarial training of shallow networks with early stopping. Unfortunately, all of the aforementioned works are limited to the lazy regime (aka, the NTK setting) which has been shown to be at odds with adversarial robustness (Wang et al., 2022). Mianjy and Arora (2022) were the first to

provide both convergence and generalization guarantees beyond the NTK regime, yet their analysis was restricted to robust realizable data distributions.

Our work stands out from prior work in several ways. First, we study the standard adversarial training algorithm commonly used in practice. Second, we do not make restrictive assumptions regarding data separability; our generative model allows for the data to be non-separable. Finally, our results hold for neural networks of arbitrary width and can be trained for arbitrary many iterations allowing $\|W^t\|$ to go to infinity, i.e., beyond the NTK regime. The following result shows that for certain step sizes and initialization, the neural network weights move far from the initialization after the first step of adversarial training based on gradient descent.

**Proposition 3.5.** Consider the same setting as in Theorem 3.1. Then, for some absolute constant $C > 1$, with probability at least $1 - 2\delta$ over the random initialization and the draw of an i.i.d. sample, we have that $\frac{\|W^1 - W^0\|_F}{\|W^0\|_F} \geq \frac{\gamma(199\|\mu\| - 200\alpha)}{1000}$.

Finally, we note that Dan et al. (2020) establish a minimax-type lower bound for the classification excess risk in the conditional Gaussian model, with a bound of $\Omega_P\left(\exp\left(-\left(\frac{1}{8} + o(1)\right)r^2\right)\frac{d}{n}\right)$[1], where $r$ is the adversarial signal-to-noise ratio; this bound is shown to be achieved by a plug-in linear estimator. While useful, their result does not elucidate why adversarial training helps train robust networks. It also remains to be seen if adversarial training can achieve a matching upper bound.

## 4 PROOF SKETCH

We begin by providing some intuition for our proof. We show that when the perturbation size is relatively small, the trajectory of the adversarial training remains close to that of the standard training. Furthermore, given a good initialization of the neural network the dynamics of the training algorithm can be shown to be nearly linear. We also leverage a result from high dimensional probability, that the training data we draw is (nearly) separable even though the underlying data distribution is non-separable. We show that both of these events happen with high probability and establish what we refer to as a "good" run of the algorithm and are central to our proof.

Next, we formalize this intuition and provide a brief proof sketch of our main result. We focus primarily on neural networks with smooth activation function (i.e., Theorem 3.1) and note the differences in the analysis when extending the result to the non-smooth activation functions. In our analysis, we borrow many ideas from Frei et al. (2022) and Xu and Gu (2023). However, the extension is not straightforward and our focus in this section is on highlighting the technical challenges we overcome and the key insights we utilized in our analysis. For detailed proofs, we refer the reader to the Appendix.

### 4.1 GENERALIZATION GUARANTEE

As a proof strategy we seek to get an upper bound on the robust test error in terms of a lower bound on the normalized expected conditional margin. This follows using a concentration argument given that $\mathcal{D}_{\text{clust}}$ is $\lambda$-strongly log-concave.

**Lemma 4.1.** Suppose that $\mathbb{E}_{(x,y_c)\sim\mathcal{D}_c}[y_c f(x;W)|y_c = \bar{y}] - \|W\|_2 \alpha \geq 0$ holds for both $\bar{y} = 1$ and $\bar{y} = -1$. Then, there exists a universal constant $c > 0$ such that

$$L_{\text{rob}}^{0/1}(W) \leq \beta + \sum_{\bar{y}\in\{-1,+1\}} \exp\left(-c\lambda\left(\frac{\mathbb{E}_{(x,y_c)\sim\mathcal{D}_c}[y_c f(x;W)|y_c = \bar{y}]}{\|W\|_2} - \alpha\right)^2\right)$$

Next, we need to show that the assumption in Lemma 4.1 does indeed hold for our setting. Here, we leverage the smoothness property of the activation function to derive a lower bound on the increment in the un-normalized margin for an independent test example $(x, y)$.

**Lemma 4.2 (Informal).** For some constant $C_2$, with high probability, we have for any $t \geq 0$ and $(x, y) \in \mathbb{R}^d \times \{\pm 1\}$, there exist $\tilde{\rho}_i^t = \rho\left(W^t, \tilde{x}_i^t, x\right) \in [\gamma^2, 1]$ such that

$$y\left[f(x;W^{t+1}) - f(x;W^t)\right] \geq \frac{\eta}{n}\sum_{i=1}^n \tilde{g}_i(W^t)\left(\tilde{\xi}_i^t \left\langle y_i\tilde{x}_i^t, yx\right\rangle - \frac{H\|x\|^2 C_2^2 d\eta}{2\sqrt{m}n}\right).$$

---

[1]For a sequence of random variables, $X_n$, and corresponding constants $c_n$, $X_n = \Omega_P(c_n)$ denotes that $c_n/X_n$ converges to zero in probability as $n \to \infty$.

where $\tilde{g}_i(\mathrm{W}^t) = -\ell'(y_i f(\tilde{\mathrm{x}}_i^t; \mathrm{W}^t)) = 1/(1 + \exp\left(y_i f(\tilde{\mathrm{x}}_i^t; \mathrm{W}^t)\right))$.

For the non-smooth activation function, we get a similar result which we defer to the Appendix due to space constraints. Finally, we seek a positive lower bound on un-normalized expected conditional margin for model $\mathrm{W}^t$ by expressing it in terms of the cumulative increments of margin; i.e., showing $\mathbb{E}_{(\mathrm{x},y_c) \sim \mathcal{D}_c | y_c = 1}[y_c f(\mathrm{x}; \mathrm{W}^t)] = \sum_{t=1}^{T} \mathbb{E}_{(\mathrm{x},y_c) \sim \mathcal{D}_c | y_c = 1}[y_c f(\mathrm{x}; \mathrm{W}^t) - y_c f(\mathrm{x}; \mathrm{W}^{t-1})] + \mathbb{E}_{(\mathrm{x},y_c) \sim \mathcal{D}_c | y_c = 1}[y_c f(\mathrm{x}; \mathrm{W}^0)]$. A positive lower bound holds trivially positive if $\langle y_i \tilde{\mathrm{x}}_i^t, y_c \mathrm{x} \rangle$ is always bounded below by some positive constant. However, due to the presence of noisy labels $y_i$ and adversarial examples $\tilde{\mathrm{x}}_i$, $\langle y_i \tilde{\mathrm{x}}_i^t, y_c \mathrm{x} \rangle$ may be negative. Note, though, that the term $\langle y_i \tilde{\mathrm{x}}_i^t, y_c \mathrm{x} \rangle$ scales with $\tilde{g}_i(\mathrm{W}^t)$. If we can show that $\tilde{g}_i(\mathrm{W}^t)$ is of the same order across all training examples, and assume a small perturbation budget and that only a small fraction of labels are noisy, then we can mitigate the effect of the negative terms. The key lemma providing such a result by bounding the loss ratio is as follows.

**Lemma 4.3** (Informal). Given Assumption 1, there is an absolute constant $C_r > 0$ such that with high probability, we have for all $t \geq 0$, $\max_{i,j \in [n]} \frac{\tilde{g}_i(\mathrm{W}^t)}{\tilde{g}_j(\mathrm{W}^t)} \leq C_r$.

To see why the above holds, note that for any given $i, j \in [n]$, we have that $\frac{\tilde{g}_i(\mathrm{W}^t)}{\tilde{g}_j(\mathrm{W}^t)} \leq \max\left\{2, \frac{2\exp\left(-y_i f(\tilde{\mathrm{x}}_i^t; \mathrm{W}^t)\right)}{\exp\left(-y_j f(\tilde{\mathrm{x}}_j^t; \mathrm{W}^t)\right)}\right\}$, where $\tilde{\mathrm{x}}_i^t = \arg\min_{\tilde{\mathrm{x}}_i \in \mathcal{B}_2(\mathrm{x}_i; \alpha)} y_i f(\tilde{\mathrm{x}}_i; \mathrm{W}^t)$. For successive iterates we get that $\frac{\exp\left(-y_i f(\tilde{\mathrm{x}}_i^{t+1}; \mathrm{W}^{t+1})\right)}{\exp\left(-y_j f(\tilde{\mathrm{x}}_j^{t+1}; \mathrm{W}^{t+1})\right)} \leq \frac{\exp\left(-y_i f(\tilde{\mathrm{x}}_i^t; \mathrm{W}^t)\right)}{\exp\left(-y_j f(\tilde{\mathrm{x}}_j^t; \mathrm{W}^t)\right)} \cdot \frac{\exp\left(y_i f(\tilde{\mathrm{x}}_i^{t+1}; \mathrm{W}^t) - y_i f(\tilde{\mathrm{x}}_i^{t+1}; \mathrm{W}^{t+1})\right)}{\exp\left(y_j f(\tilde{\mathrm{x}}_j^t; \mathrm{W}^t) - y_j f(\tilde{\mathrm{x}}_j^t; \mathrm{W}^{t+1})\right)}$. Finally, we use induction to complete the proof.

For smooth activation function, the proof of Lemmas 4.2 and 4.3, follows by controling the term $y\left[f(\mathrm{x}; \mathrm{W}^{t+1}) - f(\mathrm{x}; \mathrm{W}^t)\right]$ via Taylor approximation. For non-smooth activation functions, we need to ensure that there exist enough neurons have positive activations at initialization.

**Remark 4.4.** We can modify Assumption 2 by allowing the network initialization to depend on the training data: $a_s \mathrm{w}_s^0 = \frac{\hat{\mu}}{\|\hat{\mu}\|} \omega_{\mathrm{init}} \sqrt{d}$ where $\hat{\mu} = \frac{1}{n} \sum_{i=1}^{n} y_i \mathrm{x}_i$. Then, Assumption (B8) can be relaxed to allow $\alpha \leq \mathcal{O}(\|\mu\|)$. Under Assumption 2 with the above modifications, Lemma 4.5 is still applicable and therefore Theorem 3.3 continues to hold.

**Lemma 4.5** (Informal). Given Assumption 2, with high probability, for all $s \in [m]$, we have $\left|\left\{i \in [n] : y_i = a_s, \langle \mathrm{w}_s^0, \mathrm{x}_i \rangle \geq \alpha \left\|\mathrm{w}_s^0\right\|\right\}\right| = \Theta(n)$; for all $i \in [n]$, we have $\left|\left\{s \in [m] : y_i = a_s, \langle \mathrm{w}_s^0, \mathrm{x}_i \rangle \geq \alpha \left\|\mathrm{w}_s^0\right\|\right\}\right| = \Theta(m)$.

We further show that the number of positive neurons remains large throughout the training process.

### 4.2 CONVERGENCE GUARANTEE

In order to control the robust training loss, a naive approach would be to decouple the increment of the robust training loss, from iterate $t$ to $t + 1$, into two terms as follows:

$$\widehat{L}_{\mathrm{rob}}(\mathrm{W}^{t+1}) - \widehat{L}_{\mathrm{rob}}(\mathrm{W}^t)$$
$$= \frac{1}{n} \sum_{i=1}^{n} \left[\left(\ell(y_i f(\tilde{\mathrm{x}}_i^{t+1}; \mathrm{W}^{t+1})) - \ell(y_i f(\tilde{\mathrm{x}}_i^t; \mathrm{W}^{t+1}))\right) + \left(\ell(y_i f(\tilde{\mathrm{x}}_i^t; \mathrm{W}^{t+1})) - \ell(y_i f(\tilde{\mathrm{x}}_i^t; \mathrm{W}^t))\right)\right].$$

The second term can be controlled by the smoothness property of the loss function. The first term, unfortunately, is upper bounded by $\left\|\mathrm{W}^{t+1}\right\| \left\|\tilde{\mathrm{x}}_i^{t+1} - \tilde{\mathrm{x}}_i^t\right\|$, and the robust training loss hence inevitably depends on the norm of iterates $\left\|\mathrm{W}^{t+1}\right\|$ if no additional assumptions are made. This poses a problem if we do not constrain the model weights within a bounded domain, as $\left\|\mathrm{W}^t\right\|$ may tend to infinity as the number of epochs increases. To mitigate this issue, we instead control the robust training loss via the norm of the iterates. Specifically, we first show that $\widehat{L}_{\mathrm{rob}}(\mathrm{W}^T) \leq \frac{2}{T} \sum_{t=0}^{T-1} G_{\mathrm{rob}}(\mathrm{W}^t)$ where $G_{\mathrm{rob}}(\mathrm{W}) := \frac{1}{n} \sum_{i=1}^{n} \max_{\tilde{\mathrm{x}}_i \in \mathcal{B}_2(\mathrm{x}_i, \alpha)} -\ell'(y_i f(\tilde{\mathrm{x}}_i; \mathrm{W}))$; this holds due to a property of the loss $\ell(\cdot)$ (see the Appendix for more details). We then bound $G_{\mathrm{rob}}(\mathrm{W}^t)$ by a constant scaling of $\left\langle -\nabla \widehat{L}_{\mathrm{rob}}(\mathrm{W}^t), \mathrm{V} \right\rangle$, where $\mathrm{V} \in \mathbb{R}^{m \times d}$ is a matrix with row $\mathrm{v}_s = a_s \mu / \|\mu\|$. We achieve this result using Lemma 4.3 and the fact that only a small fraction labels are noisy. Given $\sum_{t=0}^{T-1} \left\langle -\nabla \widehat{L}_{\mathrm{rob}}(\mathrm{W}^t), \mathrm{V} \right\rangle = \left\langle \mathrm{W}^T, \mathrm{V} \right\rangle - \left\langle \mathrm{W}^0, \mathrm{V} \right\rangle \leq \left\|\mathrm{W}^T\right\|_F + \left\|\mathrm{W}^0\right\|$, the only thing we need to

prove is that the growth rate of $\|W_T\|$ is smaller than $\mathcal{O}(T)$. This property holds for both smooth activation functions that satisfy our construction and non-smooth activation functions such as ReLU and Leaky ReLU.

## 5 EXPERIMENTS

In this section, we present a simple empirical study on a synthetic dataset to support our theoretical results. We follow the generative model in Section 2 to synthesize a dataset with independent label flips when generating $y$ from $y_c$. We set $\mu = \|\mu\|_2 [1, 0, 0, \ldots, 0]^\top$, $\beta = 0.1$, and generate $n = 100$ training samples and 2K test samples with the noise vector sampled from the standard multivariate Gaussian distribution, $\xi \sim \mathcal{N}(0, I)$. We train a two-layer ReLU network with width 1K. We use the default initialization in PyTorch and train the network applying full-batch gradient-descent based adversarial training using logistic loss for 1K iterations. We use PGD attack to generate adversarial examples with attack strength $\alpha/\|\mu\|$ and attack stepsize $\alpha/5\|\mu\|$ for 20 iterations. The outer minimization is trained using an initial learning rate of 0.1 with decay by 10 after training for every 500 iterations. We note that adversarial training achieves 100% robust training accuracy. We estimate the robust test accuracy using the same PGD attack. We consider settings with varying dimension $d$ and attack strength $\frac{\alpha}{\|\mu\|}$.

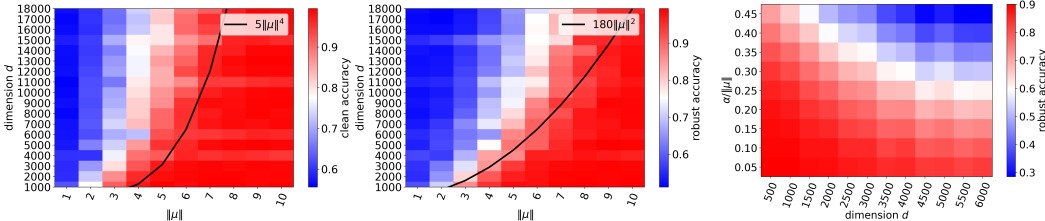

Figure 1: Clean test accuracy (left) / robust test accuracy (right) as a function of signal size $\|\mu\|$ and dimension $d$, for a fixed perturbation ratio $\alpha/\|\mu\| = 0.1$.

Figure 2: Robust test accuracy as a function of $d$ and $\frac{\alpha}{\|\mu\|}$ for a fixed $\|\mu\| = 5$.

For our first experiment, we fix the perturbation ratio $\frac{\alpha}{\|\mu\|} = 0.1$, and vary the value of the signal strength $\|\mu\|$ from 1 to 10 and the dimension $d$ from 1K to 18K. We show the results in Figure 1 as a heat map of clean accuracy and robust accuracy averaged over ten independent random runs. We observe a phase transition for both clean accuracy and robust accuracy at the value of dimension $d$ around $\mathcal{O}(\|\mu\|^4)$ for clean accuracy and $\mathcal{O}(\|\mu\|^2)$ for robust accuracy. This is consistent with the main theorems (see discussion in Section 3.3).

For our next experiment, we fix the signal size $\|\mu\| = 5.0$, vary dimension $d$ from 500 to 6K and perturbation ratio $\frac{\alpha}{\|\mu\|}$ from 0.05 to 0.45. Figure 2 plots the robust accuracy as a heat map averaged over ten independent runs. Our findings indicate that, increasing the dimension leads to a smaller perturbation ratio required to achieve the same level of robust test accuracy.

We observe the same trends on the MNIST dataset even though the data generative assumptions are no longer valid. We defer a detailed discussion of experiments on MNIST to the Appendix.

## 6 CONCLUSION

We presented the convergence and generalization guarantees for adversarial training of two-layer neural networks of arbitrary width under a non-separable data distribution. Our work suggests several promising future directions. Our results assume a generative model with a structured log-concave data distribution. It is natural to explore whether our findings can be extended to more general data distributions. Another interesting direction is to investigate whether our results generalize to the setting where the data dimension and the number of training samples have the same scale. Finally, we note that our main result only partially characterizes the phase transition from small to large test errors for small and large attack strengths, respectively. An important next step is to provide generalization guarantees for attacks of moderate strength and to explore the relationship between the perturbation size, signal size, dimension, and the number of training samples.

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
