# OpenReview forum: "Robustness Guarantees for Adversarial Training on Non-Separable Data"
_ICLR.cc/2024/Conference — Submitted to ICLR 2024_

### Official Review · Reviewer_4oeH · 2023-10-16

**Soundness:** 3 good
**Presentation:** 3 good
**Contribution:** 3 good
**Rating:** 6
**Confidence:** 3

**Summary:**

This paper provides convergence and generalization guarantees for adversarial training of two-layer neural networks on non-separable data, for both smooth and non-smooth activate functions. The experimental results are consistent with the theory.

**Strengths:**

- This paper takes a great step toward understanding adversarial training (non-smooth activate functions, non-separable data, and goes beyond the NTK regime)

**Weaknesses:**

- The model is a two-layer neural network with the weights for the second layer fixed, which is too simple compared with DNNs. And the assumption about the data is too simple.
- It seems that the paper [1] is very relevant to this work, detailed discussion about the similarities, differences, and the superiority of this paper should be added.

[1] Feature Purification: How Adversarial Training Performs Robust Deep Learning.

**Questions:**

See weaknesses.

---

> ### Author Response · Authors · 2023-11-16
> **Response to Reviewer 4oeH**
>
> We thank the reviewer for pointing our the related references and will incorporate them into the final manuscript.

---

### Official Review · Reviewer_jW96 · 2023-10-28

**Soundness:** 3 good
**Presentation:** 3 good
**Contribution:** 3 good
**Rating:** 6
**Confidence:** 3

**Summary:**

The authors consider adversarial training for neural networks of one hidden layer and prove that, under certain assumptions, it converges to an arbitrarily small robust loss. An important aspect of the theoretical results of the paper is that they hold  for NN of finite width. The results are empirically investigated on an example with a synthetic dataset.

**Strengths:**

- Providing convergence guarantees for adversarial training of neural networks is certainly a problem of interest for the ICLR community

- In contrast with the existing literature, which mostly focus on providing theoretical results in the infinite-width regime, the results of the authors also hold for neural networks of finite width. I see this as an important contribution

- The paper is well written

**Weaknesses:**

- My main issue with the submission is that the assumptions made by the authors are quite restrictive for most applications. For instance, the assumption that the number of training data is much lower than the dimension of the input space, but still larger than C log(1/\delta) is restrictive for many applications where neural networks are employed and where the amount of data is generally larger than the input dimension.

- Experimental results are only limited to a synthetic example. The authors also report some experiments on MNIST on the Appendix. Why not reporting them in the main text?

**Questions:**

- Where the constant C in Assumption 1 comes from? Can you bound it for some specific applications?

- Why in the experiments in Figure 2 robust accuracy seems to decrease with increasing the dimension d for a fixed perturbation ratio? Is this in line with your theoretical results?

Minor:

- why do you need to overload the notation for the l_2 norm in the first line of Page 3. Can't you simply use always || \cdot || ?

---

> ### Author Response · Authors · 2023-11-16
> **Response to Reviewer jW96**
>
> **[W1]**
>
> High dimensional settings, wherein $d \gg n$, are common in applications of machine learning; e.g., bioinformatics, computer vision applications involving video/multimedia data, natural language processing, etc. The requirement that $n$ is larger than a constant, i.e., $n > C \log(1/\delta)$ is also very mild. Furthermore, we note that while we limit ourselves to these settings in order to build a theory, in application we may see that our results extend beyond this setting. Also, this paves a path for future work to explore which assumptions are necessary and which are not.
>
>
> **[W2]**
>
> We report MNIST in the Appendix due to space limitations. We can switch them if the reviewer prefers that.
>
> **[Q1]**
>
> In Section 3.1, $C$ depends on various problem parameters including $\gamma, H, c_1,c_2,\zeta$ and $\kappa$.
>
> **[Q2]**
>
> Yes, this empirical result is in line with our theorem. Higher dimension leads to a lower robust accuracy as per our result, since larger $d$ leads to a larger upper bound on the robust test error.
>
> **[Q3]**
> Yes we can simply use $\| \cdot \|$.

---

### Official Review · Reviewer_GnZf · 2023-10-29

**Soundness:** 3 good
**Presentation:** 3 good
**Contribution:** 2 fair
**Rating:** 3
**Confidence:** 4

**Summary:**

This paper provides a theoretical analysis of adversarial training. Specifically, it establishes the convergence guarantees for adversarial training of a specific two-layer neural network and provides generalization guarantees for both the clean test error and the robust test error. Additionally, the paper conducts experiments to validate the theoretical results.

**Strengths:**

Compared to previous theoretical work on adversarial training, this paper does not require some strong assumptions such as linear separability or lazy training.

**Weaknesses:**

1: The setting, proof approach, and techniques in this paper completely follow the previous works [1, 2] for benign overfitting, but this is a high-dimensional data setting that is different from the typical environment of robustness problems.

2: The technical difficulty of this paper is mainly the processing of the adversarial part. However, since the attack intensity $\alpha$ needs to take a very small value, such an expansion does not have great technical difficulty and contribution.

3: While the authors claim that the results in this paper are applicable to any width, for overparameterized networks which $m\gg n, d$, the results presented may lack significance.

4: The discussion of overfitting with adversarial training is unconvincing. Such a discussion can only show that the results of this paper are not inconsistent with the phenomenon in [3], but it still cannot provide a reasonable explanation for the phenomenon in [3].

**Questions:**

I believe that in the setting of this paper, which is similar to the [1, 2] with a relatively small $\alpha$, one can potentially obtain a similar bound for robust error even without using adversarial training, by using standard SGD. I wonder what the author's perspective on this is.

[1] Spencer Frei, Niladri S Chatterji, and Peter Bartlett. Benign overfitting without linearity: Neural network classifiers trained by gradient descent for noisy linear data. In Conference on Learning Theory, pages 2668–2703. PMLR, 2022.

[2] Xingyu Xu and Yuantao Gu. Benign overfitting of non-smooth neural networks beyond lazy training. International Conference on Artificial Intelligence and Statistics, pages 11094–11117, 2023.

[3] Leslie Rice, Eric Wong, and Zico Kolter. Overfitting in adversarially robust deep learning. In International Conference on Machine Learning, pages 8093–8104. PMLR, 2020.

---

> ### Author Response · Authors · 2023-11-16
> **Response to Reviewer GnZf**
>
> **[W1, W2]**
>
> **Regarding technical difficulty and contribution.**
>
> Adding the adversarial perturbations requires certain ingenuity for the proofs to go through; they are not straightforward as you can check details from the Appendix. We here point out the contributions
> 1. We fix mistakes in the proofs
> In prior work of Frei et al. (2022) and Xu and Gu (2023). Please see **discussion in Appendix Section B.1 first paragraph** for details on technical improvements over Frei et al. (2022), and Prop. B.12-Lemma B.15 for a more rigorous concentration argument over Xu and Gu (2023).
> 2. The proof of our convergence guarantee also differs from the original -- our Lemma B.11 has a different proof strategy than Lemma 14 in Frei et al. (2022). Following the previous work to give a result in an adversarial setting leads to a vacuous bound. Please see the beginning of Section 4.2.
> 3. We establish a robust test error lower bound and explore its relationship with the upper bound, suggesting an almost tight bound under certain conditions. These are interesting findings not present in previous literature.
>
>
>
>
>
> **Regarding the perturbation size.**
>
> We disagree with the reviewer's comment that ``intensity $\alpha$ needs to take a very small value". For smooth activation function, the perturbation size can be as large as $\alpha=O(\|\mu\|)$ when $d=\Theta(n\|\mu\|^2)$.
>
>
>
> **Regarding high dimensionality data in robust setting.**
>
> Most ML problems are high dimensional. The ML community has increasingly been interested in theoretically understanding high-dimensional data under robust learning framework; see Shafahi et al. (2019), Mahloujifar et al. (2019), Richardson and Yair (2021), for example.
>
> $$$$
>
> **[W3]**
>
> **Regarding any width networks.**
>
>  We respectfully disagree with the reviewer's comment. Our guarantees hold for networks of any width. This does not mean that our guarantees become insignficant in overparamterized settings. Furthermore, the convergence and generalization guarantees for overparametrized networks in the lazy regime or the Neural Tangent Kernel (NTK) regime (which is the dominant framework in theoretical deep learning), **do not extend to the adversarial setting.** The work of Wang et al. (2022) shows that networks trained in this regime are not robust. Our work shows that adversarial training actually returns networks that are outside the lazy regime (see Proposition 3.5) and are guaranteed to generalize adversarially robustly.
>
> $$$$
>
> **[W4]**
>
> **Regarding overfitting discussion.**
>
> Nowhere in our discussion do we claim to explain the robust overfitting phenomenon observed in Rice et al. (2020). We simply remark that our results are not in conflict with their findings -- that remark is in no way trying to explain any phenomenon.
>
> $$$$
>
> **[Q]**
>
> That is a good question. We are not sure if the result will hold for SGD, but for GD it is possible that we can give Lemma 4.3, but with a different bound. Note though, that is only an intermediate step and we have not worked out all the details.
>
> $$$$
>
> Reference:
>
> [1] Frei et al. Benign overfitting without linearity: Neural network classifiers trained by gradient descent for noisy linear data. In Conference on Learning Theory, 2022.
>
> [2] Xu and Gu. Benign overfitting of non-smooth neural networks beyond lazy training. International Conference on Artificial Intelligence and Statistics, 2023.
>
> [3] Shafahi et al. "Are adversarial examples inevitable?." In Seventh International Conference on Learning Representations, 2019.
>
> [4] Mahloujifar et al. "The curse of concentration in robust learning: Evasion and poisoning attacks from concentration of measure."Proceedings of the AAAI Conference on Artificial Intelligence. Vol. 33. No. 01. 2019.
>
> [5] Richardson and Yair. "A bayes-optimal view on adversarial examples."The Journal of Machine Learning Research 22.1 (2021): 10076-10103.
>
> [6] Rice et al. Overfitting in adversarially robust deep learning. In International Conference on Machine Learning, PMLR, 2020.
>
> [7] Wang et al. "Adversarial robustness is at odds with lazy training."Advances in Neural Information Processing Systems, 2022.

---

> ### Comment · Reviewer_GnZf · 2023-11-20
>
> Dear Authors,
>
> Thank you for the response. I still have some concerns about this paper:
>
> 1: Regarding the perturbation size:
>
> In my understanding,  $2\times \mu$ actually describes the separation distance between the 'centers' of two different classes. Therefore, for Section 3.1, the assumption $\alpha \leq 0.99 \mu$ does not represent a very large perturbation size. And, in Section 3.2, $\mu$ is large, and $\alpha \leq C$ represents a very small perturbation size.
>
> 2: Regarding any width networks:
>
> Based on assumptions B.3 and B.7, when the network width tends to infinity, both the number of data n and data dimensions d should also tend to infinity for Theorem 3.3 to hold. That is to say, for fixed n and d, Theorem 3.3 does not hold for any width.
>
> 3: Regarding "lazy training":
>
> I cannot agree with the author's response: "Our work shows that adversarial training actually returns networks that are outside the lazy regime (see Proposition 3.5) and are guaranteed to generalize adversarially robustly."
>
> The relationship between the training regime of neural networks and their initialization has been studied [1]. A non-lazy training regime is not induced by adversarial training; without adversarial training, neural network training is non-lazy when the network initialization is sufficiently small, some papers with similar settings also demonstrated similar non-lazy training results without using AT [2]. In fact, there have been experiments indicating that adversarial training becomes 'lazy' earlier than standard training in a lazy training regime [3].
>
> 4: Regarding my raised question:
>
>
> I think the authors did not provide a satisfactory answer. Directly setting $\alpha = 0$ (in training but remaining in robust loss) in the Thm 3.1 and Thm 3.3 would result in the degradation of the results to simple gradient descent. And, the bounds for Theorems 3.1 and 3.3 in this paper are optimal at $\alpha = 0$ (Because the $\alpha$ in robust loss only affects the last $\alpha$ that appears in the Thm 3.1). This implies that under the settings of this paper, a good robust loss can be attained using gradient descent without adversarial training (and potentially even better than adversarial training). This seems contradictory to the motivation of this paper.
>
>
> Thank you,
>
> Reviewer GnZf
>
> Reference:
>
> [1] Tao Luo, et al. Phase diagram for two-layer relu neural networks at infinite-width limit. JMLR, 2021.
>
> [2] Spencer Frei, et al. Benign Overfitting without Linearity: Neural Network Classifiers Trained by Gradient Descent for Noisy Linear Data. COLT 2022.
>
> [3] Nikolaos Tsilivis, et al. What Can the Neural Tangent Kernel Tell Us About Adversarial Robustness? NeurIPS 2022.

---

> ### Author Response · Authors · 2023-11-21
>
> Thank you for the discussion.
>
> *Regarding the perturbation size and networks of any width:*
>
> Below we discuss smooth activation function and non-smooth activation function separately. Please remember that the analysis for non-smooth activation function is more challenging.
>
> For smooth activation functions:
>
> 1. The best achievable condition is $\alpha \leq 0.99||\mu||$; $\alpha > ||\mu||$ results in a robust test error of at least $0.5$ (refer to Theorem 3.4). This is not surprising because even for linearly separable data, if the margin is $\mu$ and class conditionals (i.e., $p(x|y)$) are supported on (marginal) hyper-planes, i.e., at a distance of $\mu$ from each other, then a perturbation of more than $\mu$ will result in error of $0.5.$
>
> 2. We do not require any assumptions on width for this part.
>
> For non-smooth activation functions:
>
> 1. First of, we do discuss in paper how we can relax this assumption also to $\alpha\leq 0.99\|\mu\|$ if we select a data-dependent initialization. See Remark 4.4 and Remark B.16. But, note that in literature, we are most concerned about settings where the adversarial perturbations are so small, that they are nearly imperceptible. This is why we consider $\alpha\leq \sqrt{n/d}||\mu||$ (or $\alpha\leq$ constant) which is reasonable in high-dimensional setting. Again, existing literature, e.g., [1,2,3], focuses on understanding the robustness guarantees with perturbation of size $O(||x||/\sqrt{d})$.
>
> 2. We note that what we mean when we say that ``our results hold for networks of any width'' is that we do not require any strong assumptions (e.g., extreme overparametrization) on the network. This is standard usage to characterize results of this type and very typical in nearly all papers in theory of deep learning.
> We only require very mild assumptions on width, specifically $\Omega(\log(n/\delta)) \leq m \leq O(\exp(n)\delta)$. These requirements are also found in [4].
>
> [1] Bartlett et al. "Adversarial examples in multi-layer random relu networks."Advances in Neural Information Processing Systems 34 (2021): 9241-9252.
>
> [2] Montanari and Wu. "Adversarial examples in random neural networks with general activations." Mathematical Statistics and Learning 6.1 (2023): 143-200.
>
> [3] Wang et al. "Adversarial robustness is at odds with lazy training."Advances in Neural Information Processing Systems, 2022.
>
> [4] Xu and Gu. Benign overfitting of non-smooth neural networks beyond lazy training. International Conference on Artificial Intelligence and Statistics, 2023.
>
>
>
> $     $
>
>
>
>
> *Regarding "lazy training":*
>
> **Nowhere** in our paper or in our response, do we claim that a non-lazy regime is ONLY induced by adversarial training. All we are saying is that our analysis shows that for the setting we consider, adversarial training returns networks that are outside the lazy regime and, therefore, our results are not in contradiction with previous results that suggest that lazy regime is at odds with robustness. We feel that our comment was twisted badly here for the sake of a baseless argument. In any case, our result does not contradict with [1], and we are happy to include [1] in our related work discussion.
>
> [1] Tao Luo, et al. Phase diagram for two-layer relu neural networks at infinite-width limit. JMLR, 2021.
>
>
> $     $
>
>
>
>
> *Regarding the question.*
>
> If $\alpha=0$, the L.H.S. of the bound is no longer robust loss.
>
> Regardless, we agree that there exist a similar form (albeit with different constants in the bound) of robust generalization guarantees for neural network trained non-adversarially with gradient descent. We want to point out that this observation doesn't contradict the motivation of our work. We would like to emphasize again that our results serve as the first step towards understanding the adversarial trained neural network without linearly separable assumption and beyond lazy training regime.

---

> > ### Comment · Reviewer_GnZf · 2023-11-22
> >
> > Dear Authors,
> >
> > Firstly, I'm glad that the author agrees with my viewpoint: "There exists a similar form of robust generalization guarantees for a neural network trained non-adversarially with gradient descent." and "the results in this paper regarding non-smooth activation functions are not entirely applicable to networks of any width". For the author's other responses, I have the following supplements:
> >
> > Regarding the comparison between adversarial training and non-adversarial training with gradient descent:
> >
> > In the author's response, there's a statement: "If $\alpha = 0$, the L.H.S. of the bound is no longer robust loss." So, I want to clarify a potential misunderstanding the author might have from my previous response.
> >
> > Consider Theorem 3.1, there are three $\alpha$. In the order of appearance, the first two $\alpha$ are associated with $\alpha$ in adversarial training (Algorithm 1), denoted as $\alpha_1$, while the third $\alpha$ is associated with $\alpha$ in the robust loss, denoted as $\alpha_2$ (as $\alpha_1$ and $\alpha_2$ are mutually independent due to the separation of the robust loss impact in the first step of the proof in the paper). As I mentioned earlier, directly setting $\alpha_1 = 0$ yields a bound with better constants compared to adversarial training. This setting corresponds to non-adversarial training with the same robust loss analyzed in this paper.
> >
> > I don't deny that the author provides new bounds for adversarial training. However, both from the results and technically, this bound is derived from adding small perturbations under non-adversarial training (even neglecting almost all adversarial directions and directly using random perturbations, resulting in worse constants than non-adversarial training). In my view, this doesn't effectively inform our understanding of adversarial training.
> >
> > Regarding lazy training:
> >
> > My previous response explicitly stated my disagreement with the statement in the response section, where the comparison between "the NTK network of non-adversarial training is lazy" and "the non-laziness of networks trained by adversarial training in the specific setting of this paper" without further elaboration. Such a statement in a discussion section of a paper focused on adversarial training will create ambiguity for readers. As I mentioned earlier, the non-lazy training regime of the network in the setting of this paper is a natural consequence and should be seen as a supplementary clarification to the main theorem of the paper.
> >
> > Moreover, I disagree with this statement in the author's newest response: "Our results are not in contradiction with previous results that suggest that lazy regime is at odds with robustness." In fact, under the specific data setting of this paper, there exists a similar form of robust generalization guarantees for a neural network under the lazy training regime by applying similar perturbations in this paper to [1]. This is also why I mentioned in my previous response that this specific high-dimensional data setting is not a reasonable setting for studying robustness.
> >
> > In conclusion, I believe the author needs to modify the assumptions, results, and proofs of the paper to give more meaningful theoretical results, add necessary discussions, and revise inaccurate statements.
> >
> > Thank you,
> >
> > Reviewer GnZf
> >
> > Reference:
> >
> > [1] Zhenyu Zhu, et al. Benign Overfitting in Deep Neural Networks under Lazy Training. ICML, 2023.

---

> > > ### Author Response · Authors · 2023-11-22
> > >
> > > Regarding agreeing on whether a similar robustness guarantee would hold for GD or not, we would like to say we never disagreed with it, even in our initial response. We said such a bound would hold for GD (although not for SGD) but potentially with different constants. The reviewer seems to be trying to score a point here by twisting our words. He is not acknowledging the other part of our response that this does not take away from our contributions to analyzing adversarial training. We would also like to share that even for GD, it is not trivial to show a robustness guarantee — in fact, it may not even hold *very* generally. For example, even when the data is linearly separable, and the perturbation is smaller than the margin, GD is not guaranteed to return a robust linear classifier unless it also maximizes the margin.
> > >
> > > In our setting, there is only one $\alpha$; there are no three $\alpha$’s as the reviewer claims. Again, this is a wild goose chase that the reviewer is setting us out on to make a point that is baseless. The reviewer claimed earlier that simply plugging $\alpha=0$ in our Theorems would give a robustness guarantee for GD. That claim is false. Our guarantees do not yield robustness guarantees for plain GD. If the reviewer wants to analyze the setting where, during training time, we do not perturb the data but still give a bound on robust generalization error for some positive $\alpha > 0$, that would require a different analysis. For example, Lemma 4.3, which is an intermediate step in our proof would change -- something we already said in our initial response.
> > >
> > > We are glad that the reviewer agrees that "the author provides new bounds for adversarial training".
> > > We disagree with their conclusion that ``this doesn't effectively inform our understanding of adversarial training" since it is based on the flawed reasoning as we described above.
> > >
> > > $         $
> > >
> > >  Regarding "lazy training’’, we cannot find the statements that the reviewer pasted in quotes. We are sure that we never made those comments. What we wrote from the beginning is that ``Our work shows that adversarial training actually returns networks that are outside the lazy regime (see Proposition 3.5) and are guaranteed to generalize adversarially robustly.’’ There is nothing wrong with our statement. In fact, the paper that reviewer shared has no relationship with our work. The paper in question here makes a distributional assumption that there is a robust network in the lazy regime (Assumption 6 in [1]). In general settings, where such a strong assumption does not hold, we already know from the prior work of Wang et al. (2022) that every network in lazy regime is non-robust.
> > >
> > > Finally, we would also like an acknowledgment from the reviewer that we have effectively addressed their comments regarding "the perturbation size", "the width of the network’’, and "the technical difficulty and contribution’’  that they initially raised since they have not come up in the discussion after our response. We believe with our current response, we have addressed other comments as well and would like to urge the reviewer to reconsider their rating of our paper.
> > >
> > >
> > > [1] Zhenyu Zhu, et al. Benign Overfitting in Deep Neural Networks under Lazy Training. ICML, 2023.
> > >
> > > [2] Wang et al. "Adversarial robustness is at odds with lazy training."Advances in Neural Information Processing Systems, 2022.

---

> ### Comment · Reviewer_GnZf · 2023-11-23
>
> Dear Authors,
>
> Firstly, I'd like to clarify my previous responses.
>
> 1. $\alpha$:
>
> I have corrected the possible ambiguities in my previous responses. I think I was clear in my previous response regarding the origin of $\alpha$ in the results.  The author seems to be evading this issue.
>
> 2. Lemma 4.3:
>
> The proof of the lemma relies on Lemma 4.9 in [1], differing only by constants. Moreover, Lemma 4.9 in [1] itself is a non-adversarial version of this setting. So the author’s previous statement about Lemma 4.3 is actually not true.
>
> 3. Lazy training:
>
> My concern lies in this excerpt:
> "Furthermore, the convergence and generalization guarantees for overparametrized networks in the lazy regime or the Neural Tangent Kernel (NTK) regime (which is the dominant framework in theoretical deep learning), do not extend to the adversarial setting. The work of Wang et al. (2022) shows that networks trained in this regime are not robust. Our work shows that adversarial training actually returns networks that are outside the lazy regime (see Proposition 3.5) and are guaranteed to generalize adversarially robustly."
>
> And this recent response: "Our work shows that adversarial training actually returns networks that are outside the lazy regime (see Proposition 3.5) and are guaranteed to generalize adversarially robustly.’’
>
> Both statements attempt to establish a connection between adversarial training and the non-lazy regime but overlook the crucial influence of initialization in the lazy/non-lazy regime. This might mislead readers. In fact, under the settings of this paper, whether adversarial training is used or not, it will fall into the non-lazy regime. However, this is more of a formulation issue than a core concern of the paper.
>
> 4. Relation between [2] and this paper:
>
> Both this paper and Section 3 in [2] primarily draw from [1] for their data distributions. The difference lies in initialization, resulting in different training regimes. Their proof strategies are similar (follow [1]). The authors attempt to refute Assumption 6 in [2]. But Assumption 6 applies only to Section 4 in [2]. It does not affect any of the results in Section 3 in [2].
>
> 5: "The perturbation size":
>
> The author reiterated the results from the paper: for this issue, smooth activations need perturbations smaller than the distance from the data center to the boundary, while non-smooth activations need perturbations significantly smaller than this distance. I believe further discussion on whether these values are large or small is meaningless. In fact, the data distribution in this paper is considered 'linearly separable' in many references [3][4].
>
> 6: "The width of the network":
>
> The author's response stated, "We do not require any strong assumptions (e.g., extreme over parametrization) on the network," which significantly differs from the original statement: "Our results hold for networks of any width."
>
> 7: "The technical difficulty and contribution":
>
> I previously raised the issue: "even neglecting almost all adversarial directions and directly using random perturbations," and the author seems not to have responded to this.
>
> As a reviewer, I aimed to discuss and enhance the paper with the author. However, the author's responses seem to diverge from this, so I feel further discussion would be futile. Given the author's evasiveness regarding some crucial issues, I will maintain my score.
>
> Reviewer GnZf
>
> Reference:
>
> [1] Spencer Frei, et al. Benign Overfitting without Linearity: Neural Network Classifiers Trained by Gradient Descent for Noisy Linear Data. COLT 2022.
>
> [2] Zhenyu Zhu, et al. Benign Overfitting in Deep Neural Networks under Lazy Training. ICML, 2023.
>
> [3] Zhiwei Xu, et al. Benign Overfitting and Grokking in ReLU Networks for XOR Cluster Data. arXiv:2310.02541.
>
> [4] Xuran Meng, et al. Benign Overfitting in Two-Layer ReLU Convolutional Neural Networks for XOR Data. arXiv:2310.01975.

---

> ### Author Response · Authors · 2023-11-23
>
> Dear Reviewer GnZf:
>
> 1. Regarding $\alpha=0$: your claim was blatantly wrong. It was not ambiguous. You have since changed the question to something that has nothing to do with the paper. We have already said that GD would also have similar robustness guarantees. So, entertaining your question further makes no sense but to engage in this meaningless back-and-forth, which has no constructive outcome. We are not being evasive here. You are not willing to see the reason.
>
> 2. Regarding Lemma 4.3: What we have said before about Lemma 4.3 is that it is possible to give an analogue of it for GD but with different constants. This is exactly what you say here. What is it about what we said that is not true? You seem to be in an alternate reality where you repeatedly think we are saying something we are not.
>
> 3. Regarding lazy training. For the nth time, the prior work shows that networks trained in a lazy regime are not robust. We show that adversarial training in our setting yields networks that are outside the lazy regime. Nowhere in the paper or our response, even the ones you copied here, say anything you claim we say. Again, you are twisting the reality to suit your needs. Nobody is being misled by these statements. You are simply not willing to listen to reason.
>
> 4. Regarding reference [2]. Benign overfitting in a lazy regime has nothing to do with adversarial training or robustness. You are trying to make connections that are tangential at best and definitely outside the scope of the paper.
>
> 5. Regarding our data distribution being linearly separable, you are flat-out wrong again here. Not only can our class conditionals, i.e., p(x|y), overlap, but we also have non-zero probability $\beta > 0$ of flipping the labels.
>
> 6. Regarding the width. Again, as we said before, **for smooth activation functions, we make NO assumption on width**. For non-smooth activation functions, we do make a mild assumption on width $m$ of the network; we assume that $\Omega(\log(n/\delta)) < m < O(\exp(n) \delta)$. We argue that this essentially means that our results hold for networks of any width. This is standard usage in literature as many theoretical results require the networks to be sufficiently over-parametrized. You have an issue with our phrasing that our "results hold for a network of any width’’—  this is such a minutia; is it really such a big deal?
>
> 7. We have already addressed the issue of novel technical contributions, including making the reviewer aware that Lemma 4.1 and Lemma 4.11 in the prior work of Spencer Frei et al. are broken. We provide alternate arguments to give guarantees, and they are sound and rigorous.
>
> Thanks for clarifying your stand. None of the reasons you list above are central to the results or technical contributions we make in the paper. You are focusing on minutiae like "Does this result really hold for a network of any width?’’ or ``is this a comment on training in a lazy regime more generally?’’ Even if these were genuine concerns, none of this can be a basis for rejection. None of your comments justify the low rating.
>
> Best,
> Authors

---

### Official Review · Reviewer_ycei · 2023-11-01

**Soundness:** 3 good
**Presentation:** 3 good
**Contribution:** 3 good
**Rating:** 6
**Confidence:** 3

**Summary:**

This paper studies convergence and generalization guarantees of adversarial training of two-layer neural networks with arbitrary width on non-separable data. It provides theoretical guarantees on both smooth and non-smooth activation functions. For moderately large networks, the paper shows the robust test error behaves differently on different perturbation budgets. The theoretical findings are supported by experiments on both synthetic and real-world data.

**Strengths:**

The paper is well-written. References are well cited with detailed comparisons.

**Weaknesses:**

The additional assumption in Section 3, which states that $\phi'(z)z$ and $\phi(z)$ are close, is hard to grasp. Is it another notion on the Liptshitzness on the activation function? How strong is it to assume that $c_1,c_2=0$ in ReLU network, for example?

**Questions:**

See discussion in Weaknesses.

**Details Of Ethics Concerns:**

No concerns.

---

> ### Author Response · Authors · 2023-11-16
> **Response to Reviewer ycei**
>
> In Section 3.1, the additional assumption approximates the homogeneous property, which is used for proving the convergence guarantee.
> This almost homogeneous property is not very important, because it is only used to prove the bound on training error (not test error). From this assumption we showed that if the smooth loss function is close to ReLU or leaky ReLU, we can have a small training error.
>
> Another example of almost homogeneous function:
> \begin{equation}
> \phi(x)= \begin{cases}
> x &x \leq 1,\newline
> \sqrt{2x-1} &x>1. \end{cases}
> \end{equation}
>
> We are not assuming $c_1=c_2=0$ for ReLU.
> The approximate homogeneous property holds trivially for ReLU.

---

### Meta-Review · Area_Chair_oaXg · 2023-12-05

**Metareview:**

This paper studies how adversarial training helps on robustness for binary classification. Unlike prior works which often attack this problem by assuming the data are linearly separable under distribution $D_c$, this paper takes a step further which considers a data generation distribution $D$ whose TV distance to $D_c$ is at most $\beta$ and under which linear separation breaks down. The main results demonstrate that under certain regimes, the clean test error and robust test error may converge to $\beta$.

**Strengths**
- This is a well-written paper and provides good review of the literature.
- This paper provide useful guarantees on finite-width two-layer networks with certain activation functions
- The proof looks correct.
- Some numerical studies were provided to validate the theory

**Weaknesses**
- Some arguments can be misleading or over-stated. For example, the data is indeed linearly separable by the Bayes classifier and such data generation model has been studied in a few prior works such as Chatterji & Long 2021, Frei et al 2022. The paper does consider a variant of such model that allows the observed data to be generated by some distribution whose TV distance is at most $\beta$ from the true one, but this would only introduce an additive $\beta$ error in the testing error rather than posing additional technical challenges. This can be seen from both the theorems and the proofs. In fact, the data generation model is almost same as Frei et al 2022, who called the model as noisy linear data (see their paper title) and who have presented techniques on how to handle it.

- Since dealing with the non-separable data is not a major technical contribution, the main claim would better be rephrased as providing adversarial robustness guarantees via adversarial training. However, it is unclear whether adversarial training is necessary to achieve the claimed robustness, or put slightly differently, does adversarial training lead to improved robustness to plain GD. The authors responded that the improvement is only by a constant multiplicative factor; I think this cannot justify the significance of the current results.

- The last major concern on the technical contribution is, is the derived robustness guarantee sound. As pointed out by some reviewers, the parameter regime under which the theorems hold is quite limited. For example, in Theorem 3.1, the condition '$\epsilon < 1/(2n)$' is odd; this condition says the theory breaks down as soon as $n > 1/\epsilon$. Similar issues can also be found in Assumption (A3) and (B1). In addition, a closer look at the assumptions and main theorems (e.g. Theorem 3.1) quickly shows that the paper largely follows Frei et al 2022. Although authors clarified in the submission (the last complete paragraph on page 2) that the new challenges are (1) non-separable data (this is invalid argument) and (2) non-smooth activation functions, given the many restrictions imposed on the family of non-smooth functions in Section 3.2, it is unclear how generic their results in this space are; indeed, the authors failed to justify that it is nontrivial to extend Frei et al 2022 to the current network architectures.

- Other minor issues include: (1) numerical results are insufficient; (2) the abstract and the summary of contributions in introduction can be misleading in the sense that the paper does not study any two-layer networks but the wording seems to indicate otherwise; (3) assumptions are tedious and some of them are restrictive.

**Suggestions to authors**
- Authors are suggested to clarify what real technical challenges are new when considering the current data distribution model. In particular, clarify how the treatment on such model deviates the one in Frei et al 2022.

- Authors are suggested to clarify the real benefits of adversarial vs non-adversarial training.

**Justification For Why Not Higher Score:**

Authors cannot clarify the major technical novelties. The theoretical results are improved only by a constant multiplicative factor compared to non-adversarial training, which is insufficient to justify the significance.

**Justification For Why Not Lower Score:**

N/A

---

### Decision · Program_Chairs · 2024-01-16

Reject